

# Investigating the second whitefly population outbreak within a decade in the cotton growing zone of North India

Rishi Kumar[1], Satish Kumar Sain[1], Satnam Singh[2], Suneet Pandher[2], Roop Singh Meena[3], Anil Jakhar[4], Jasjinder Kaur[5], Mandeep Pathania[5], Debashis Paul[1], Prakash A.H.[6] and Prasad Y.G.[7]

[1] ICAR-Central Institute for Cotton Research, Regional Station, Sirsa, Haryana, India
[2] Punjab Agricultural University, Regional Research Station, Faridkot, Punjab, India
[3] Agriculture Research Station, SKRAU, Sriganaganagar, Rajasthan, India
[4] CCS Haryana Agricultural University, Hisar, Haryana, India
[5] Punjab Agricultural University, Regional Research Station, Bathinda, Bhatinda, Punjab, India
[6] ICAR-Central Institute for Cotton Research, Regional Station, Coimbatore, Tamilnadu, India
[7] ICAR-Central Institute for Cotton Research, Regional Station, Nagpur, Maharashtra, India

Corresponding author
Rishi Kumar,
rishipareek70@yahoo.co.in

## ABSTRACT

The whitefly, *Bemisia tabaci* (Gennadius), is a polyphagous and major pest of cotton worldwide. Both adults and nymphs of *B. tabaci* affect the crop by causing direct and indirect damage. A severe whitefly outbreak was experienced during 2015 on cotton in North India and this was followed by a profound infestation during 2022. The present research rigorously examined whether the proliferation in the whitefly population was an outbreak or the result of a multi factor resurgence. During 2015, whitefly counts remained above the economic threshold level (ETL) between 28th and 35th Standard Meteorological Week (SMW). However, during 2022 above ETL population was observed in 27th SMW and it persisted until 36th SMW. The peak incidence of the whitefly was noticed during 31st and 29th SMW in 2015 and 2022, respectively. The early pest build up in 2022 and longer persistence (≥10 weeks) over the cotton season resulted in more damage to cotton crop. Additionally, pest survillence across the zone on the farmers' fields during 2022 revealed 44.4 per cent spots (585 out of 1,317 locations) above ETL while the corresponding locations in 2015 was 57% (620 out of 1,089). Thus, in 2022 infestation was not uniform in the entire zone wherein only few blocks of Punjab, Haryana and Rajasthan states of India experienced severe infestations of the whitefly. This study reports the complex of factors including weather, delayed sowing, use of tank mixtures/ subleathal doses of insecticides, pest resurgence *etc.* that might have possibly contributed to these upsurges in whitefly on cotton in north India.

# INTRODUCTION

The whitefly, *Bemisia tabaci* Gennadius (Hemiptera: Aleyrodidae) was first reported in Greece about 125 years ago (*Gennadius, 1889*; *Takahashi, 1936*). The first report of whitefly on cotton in India comes from Pusa (Bihar) in 1905, followed by attaining the

status of a serious pest of cotton in Punjab in 1930s (*Immaraju, 1989*; *Kumar et al., 2016*). However, *B. tabaci* assumed the status of key pest of cotton only after the introduction of synthetic pyrethroids during early 1980 for the management of cotton bollworms (*Patil, Nandhihalli & Hugar, 1990*; *Dhawan, Butter & Narula, 2007*). The ability of the whitefly to survive and proliferate rapidly is attributed to its adaptability to a wide host range of more than 900 plant species (*Oliveira, Henneberry & Anderson, 2001*; *Simmons, Harrison & Ling, 2008*). Besides cotton, its infestations has also been observed on various alternate hosts like green gram (*Vigna radiata*), black gram (*Vigna mungo*), pigeon pea (*Cajanas cajan*), cluster bean (*Cyamopsis tetragonoloba*), cucurbits (*Cucurbitaceae)*, okra (*Abelmoschus esculentus*), brinjal (*Solanum melongena*), chilli (*Capsicum annuum*) *etc.* Towards the end of the cotton season, the adults migrate to other crops such as *Brassicas*, potato, tomato and several weed hosts like peelibuti (*Abutilon* sp.), kanghi buti (*Sida* sp.), puthkanda (*Achyranthes splendens*), *etc.*, for the off-season survival (*Singh & Aggarwal, 2023*). Although suitable weather conditions for growth of *B. tabaci* are 20–33 °C temperature (*Wang & Tsai, 1996*; *Muñiz & Nombela, 2001*) but for rapid population build up of this insect, hot and humid conditions are more conducive. In the north zone whitefly exposed to maximum temperature range between 26–49 °C during the cotton season. The most favorable temperature for whitefly is between 20–34 °C (ICAR-CICR, unpublished data, 2016–2019) in the north zone of India. *B.tabaci* cause damage by sucking phloem sap that results in early wilting, premature defoliation, stunted growth, that eventually leads to yield losses. Indirect damage to crop plants is caused by the whitefly-excreted honeydew, which promotes fungal growth on leaves developing into black sooty mould resulting in decreased phytosynthetic acitivity of plant and and on opened cotton bolls,this secretion reduces the quality and market value of fibre. Cotton losses due to this whitefly have been estimated to be in the range of 15–20% and sometimes up to 30% (*Kumar et al., 2020*; *Khan & Khan, 1995*).

Pest outbreaks refers to a sudden and significant increase in the population of a pest species and typically associated with favorable environmental conditions, absence of natural predators, or an abundance of suitable host leading to extensive damage to crops. On the otherhand, pest resurgence occurs when the population of a pest species increases (after a temporary decline) due to inappropriate use of pesticides, reduction of natural enemies or inappropriate nutrient application. Resurgence may occur due to many factors, but generally resurgence is "an abnormal increase in a pest population following insecticide treatment, often far exceeding the economic injury level" (*Chelliah, Heinrichs & Smith, 1984*). Many documented instances of resurgence in pest population after application of inscetcides has been reported in cotton (*Vennilla et al., 2007*; *Kerns & Gaylor, 1993*; *Mironidis et al., 2013*). Over the years, *B.tabaci* has created a niche in the cotton agroecosystems and has become a pest of a regular occurrence in all the cotton-growing areas of North-West India. The population period of this whitefly on cotton remains between 30–150 days after sowing with occurence of more than one population peaks during the season (*Tenguri, Gawande & Kumar, 2023*).

Globally severe outbreaks of *B. tabaci* were reported in Sudan and Iran (1950s), El Salvador (1961), Mexico (1962), Brazil (1968), Turkey (1974), Israel (1976), Thailand
(1978), Arizona and California, USA (1981) and Ethiopia (1984). In 1989–90 its epidemic invasion was recorded on cotton in the USA, Mexico, South America and Asia and in 1995 again in USA (Arizona). In 1994, the outbreak of *B. tabaci* was reported on cotton crop in Australia while its occurrence appeared in the country in 1959 without much economic loss (*Gunning et al., 1995*; *Carver & Reid, 1996*; *Fransmann, Lea & DeBarro, 1998*).

In the north cotton growing zone of India only a single whitefly species, *Bemisa tabaci*, predominately its biotype or cryptic species Asia-II-1 prevailed (*Mahmood et al., 2022*). The outbreak of *B. tabaci* in 2015 in north India can be considered as one of the most devasting as it affected the cotton production, productivity and as well as the area under cultivation in 2016 (Table 1). During the cotton growing season of 2022, increased whitefly infestation was recorded in many section/blocks of the North zone. This second intense population upsurge has occurred within less than a decade's since 2015. Occurrence of *B. tabaci* severe infestation on cotton may be due to the prevailing ecological conditions, impact of climate change and host plant interactions. The main aim of present study was to have an insights about the severity of whitefly incidence in cotton crop and factors associated with these upsurges. Portions of this text were previously published as part of a preprint (https://www.researchsquare.com/article/rs-3113576/v1).

## MATERIALS AND METHODS

To have an insight into infestation level and peak population of, *B. tabaci* throughout the crop season, continuous population dynamics monitoring on adults were conducted in unprotected conditions at experimental farms located in various research institutes in the North Zone (ICAR-CICR Sirsa and CCS HAU, Hisar-Haryana, Regional Research Station, PAU, Bathinda and Faridkot-Punjab, SKNRAU, ARS Sriganaganagar, Rajasthan). Similarly, surveys were also conducted at farmers' fields locations of the North zone to record *B. tabaci* incidence and locations having incidence above the economic threshold level (ETL). The locations for the study were across three cotton growing states of north India *i.e.*, Sirsa and Hisar (Haryana), Faridkot and Bathinda (Punjab) and Sriganganagar (Rajasthan). In surveys, adults counts were considered, whereas the nymphal counts recorded were not included in general and more appropriately used in population prediction studies only.

### Population dynamics under unprotected conditions

Experiments were conducted during 2012 to 2022. Cotton genotypes from Non-*Bt* and *Bt*/BG-II backgrounds were sown at five experimental farms of North zone in an area of 500 sq. meter for each genotype. Both BG-II and Non-*Bt* genotypes were selected for the study to match the cultivation practices of local growers and their compatibility with whitefly infestations. While Non-Bt genotypes remained consistent throughout the study period; however the selection of *Bt*/BG-II cotton genotypes varied in accordance with the changing cultivation patterns among growers in the region. Following the standard observation protocol, the weekly population of whitefly was sampled on various genotypes post 1 month after sowing of the crop. The population of whitefly adults and nymphal counts was recorded from three fully formed leaves (upper, middle and lower plant strata)

**Table 1  Area, and productivity of cotton north zone during 2011 to 2022.**

| Year | Area (In lakh ha) | | | | Zonal productivity (Lint kg/ha) |
|---|---|---|---|---|---|
| | Haryana | Punjab | Rajasthan | Total North Zone | |
| 2011 | 6.41 | 5.60 | 4.70 | 16.71 | 651 |
| 2012 | 6.14 | 4.80 | 4.50 | 15.44 | 705 |
| 2013 | 5.36 | 4.46 | 3.93 | 13.75 | 729 |
| 2014 | 6.48 | 4.20 | 4.87 | 15.55 | 579 |
| 2015 | 6.15 | 3.39 | 4.48 | 14.02 | 433 |
| 2016 | 5.70 | 2.85 | 4.71 | 13.26 | 590 |
| 2017 | 6.56 | 3.85 | 5.03 | 15.40 | 624 |
| 2018 | 7.08 | 2.68 | 6.29 | 16.05 | 625 |
| 2019 | 7.23 | 2.48 | 7.60 | 17.31 | 638 |
| 2020 | 7.40 | 2.52 | 8.08 | 18.00 | 572 |
| 2021 | 6.95 | 3.04 | 7.07 | 17.06 | 599 |
| 2022 | 6.47 | 2.41 | 7.77 | 16.65 | 439 |

**Note:**
Source: Committee on cotton production and consumption (COCPC), Ministry of textile, India and ministry of agriculture and farmers' welfare, India.

of 20 randomly tagged plants (four cohorts of five plants) per plot in these cultivars at weekly intervals. The mean of the weekly population was compared for population dynamics. Based on the '19 weekly' observations recorded on BG-II and HS-6 (Non-*Bt* genotype) during the season, seasonal mean was calculated to compare the severity of the incidence.

## Pest survillence across farmers' fields

The weekly surveys were conducted at farmers' fields across north zone (Punjab, Haryana and Rajasthan) wherin number of locations having infestation above the economic threshold level (ETL) were recorded. During each survey, a total of five or more locations from five different villages were monitored for sucking pests. Whitefly counts were recorded from randomly selected three leaves one each from upper, middle and lower strata per plant. The average counts of whitefly adults per three leaves was calculated and employed for comparing and assessing the level of incidence. During the roving survey detail regarding BG-II/*Bt* hybrids sampled, date of sowing, interventions applied were also recorded but not mentioned in the manuscript.

## Weather parameters

Prevailing weather parameters like maximum temperature ($T_{max}$), minimum temperature ($T_{min}$), average temperature ($T_{mean}$), morning relative humidity (RH-M), evening relative humidity (RH-E), average relative humidity ($RH_{mean}$), cumulative rainfall during the crop duration were recorded at ICAR-CICR, Regional Station, Sirsa. Here we have also discussed the relationship between seasonal mean (24–42 SMW) of weather data (recorded during 2012 to 2022) and whitefly counts per three leaves (mean of two genotypes) at ICAR-CICR, Regional Station, Sirsa location.

## Statistical analysis

Analysis of variance (ANOVA) was conducted using SAS statistical software version 9.4 (SAS Institute Inc. 2016. SAS® 9.4 Language Reference: Concepts, Sixth Edition. Cary, NC, USA). Correlation coefficients between various quantitative characters like whitefly and prevailing weather conditions were tested using appropriate statistical test. Prior to ANOVA, all the data were checked for the normality by seeing the normal distribution curve. Means of the treatments (seasonal mean of whitefly and percentage of location crossed ETL) were separated based on the Tukey's honestly significant difference test with $\alpha = 0.05$ level of significance.

## RESULTS

The incidence of whitefly recorded during 2022 was compared with whitefly incidence recorded during the year of whitefly outbreak 2015.

In Haryana, the Sirsa district has the highest area under cotton cultivation which was selected for experimental studies under unprotected conditions as well as for farmers field locations survey to assess the ground situations. The experimental data recorded under unprotected conditions during 2012, 2013 and 2014 exhibited an increase in average whitefly adult population per three leaves (Table 2) in BG-II genotype and HS-6 (non-*Bt* genotype). Peak populations of the whitefly were recorded during 30–31st Standard Meteorological Week (SMW) during 2012 and 2013, while it reached it's highest in 37th SMW. During 2015, 2016 and 2017, the whitefly surveillance clearly showed that the situations of the whitefly population during 2015 (epidemic year) and 2017 were almost at the same level (above ETL level). The seasonal mean per three leaves under unsprayed conditions recorded were 20.32, 10.1 and 16.5 in BG-II and 20.9, 12.5, and 16.4 in HS-6 during 2015, 2016 and 2017, respectively (Table 3). During 2018, 2019 and 2020 seasonal mean (average population per three leaves) was 14, 11.1 and 13.47 in BG-II and 16.3, 13.9 and 16.52 in HS-6 (N-*Bt*), respectively. The peak population in 2018, 2019 and 2020 was recorded during 29th, 31st and 38th SMW, respectively (Table 2). During 2021 and 2022 seasonal mean per three leaves recorded was 7.02 and 33.50 in BG-II, 8.98 and 38.53 in HS-6, respectively (Table 2). Peak population of whitefly was recorded from 29th SMW onwards in both BG-II and HS-6 genotypes (Fig. 1). The seasonal mean data spanning from 2012 to 2022 reported maximum whitefly population during 2022 cotton season. The seasonal means recorded during 2022 was statistically and significantly higher than 2015 and highest among the years under study.

In Punjab, Bathinda on farm experiment conducted during 2012, 2013 and 2014, reported an increasing trend in average whitefly adult population per three leaves in BG-II genotype. Peak population of whitefly in 2014 was observed during 39th SMW (Table 2) similar to Haryana locations. During 2015, 2016 and 2017 population per three leaves ranged between 3.6–34.1 (2017) in BG-II genotype whereas during 2015 and 2016 the crop was not sown. In HS-6, whitefly adult population per three leaves ranged from 0.4–28.6 (2016), 3.2–31.5 (2017) but in 2015 the data was not recorded due to the failure of HS-6 crop (Table 2). During 2015, 2016 and 2017 peak population of whitefly was recorded between 38th and 39th SMW. During 2018, 2019 and 2020 population per three leaves
**Table 2 Seasonal mean of whitefly population recorded from the experiment conducted under unsprayed conditions.**

| Year | Sirsa | | Sri Ganganagar | | Bathinda | | Total duration (in weeks) whitefly population noticed above ETL | |
|---|---|---|---|---|---|---|---|---|
| | Seasonal mean/Three leaves (24–42 SMW) | | Seasonal mean/Three leaves (24–42 SMW) | | Seasonal mean/Three leaves (24–42 SMW) | | | |
| | BG-II genotype | Non Bt genotype | BG-II genotype | Non Bt genotype | BG-II genotype | Non Bt genotype | BG-II genotype | Non Bt genotype |
| 2012 | 3.8[h] | 3.50[g] | 15.4 | 12.1 | 19.5 | – | 0 | 0 |
| 2013 | 13.4[e] | 12.9[ef] | – | 12.8 | – | 28.9 | – | 6 (26, 28–32) |
| 2014 | 15.7[cd] | 14.2[cd] | – | 53.9 | – | 45.7 | – | 8 (29–30, 34–39) |
| 2015 | 20.3[b] | 20.9[b] | – | – | – | – | 69 (29–34) | 5 (29–33) |
| 2016 | 10.1[f] | 12.5[de] | – | 17.7 | – | 9.0 | 2 (29–30) | 3 (30, 32–33) |
| 2017 | 16.5[c] | 16.4[c] | 15.8 | 19.1 | 13.8 | 3.2 | 6 (29–33,35) | 7 (29–33, 35–36) |
| 2018 | 14.0[de] | 16.3[c] | 8.9 | – | 5.2 | – | 4 (27–30) | 5 (27–30, 33) |
| 2019 | 11.1[f] | 13.9[cde] | 10.2 | 10.1 | 3.9 | 3.8 | 1 (31) | 1 (30) |
| 2020 | 13.4[e] | 16.5[c] | 35.8 | 32.1 | 3.8 | 14.1 | 5 (27, 29, 36, 38–39) | 5 (29, 36–39) |
| 2021 | 7.0[g] | 8.9[f] | 5.6 | 9.2 | 11.7 | 6.5 | 0 | 1 (29) |
| 2022 | 33.5[a] | 38.5[a] | 52.51 | 47.88 | 25.89 | 28.10 | 10 (26–34, 36–37 & 39) | 13 (27–38, 40) |

Note:
Different letters as superscripts within a column indicate significant differences ($p < 0.05$). Means were separated based on the Tukey's honestly significant difference test with $\alpha = 0.05$ level of significance. SMW, standard meteorological weeks.

**Table 3 Population range and peak activity period of whitefly population recorded based on experiment conducted under unprotected conditions.**

| Year | Sirsa | | Sri Ganganagar | | Bathinda | |
|---|---|---|---|---|---|---|
| | Population range/Three leaves (Peak population SMW) | | Population range/Three leaves (Peak population SMW) | | Population range/Three leaves (Peak population SMW) | |
| | BG-II genotype | Non Bt genotype | BG-II genotype | Non Bt genotype | BG-II genotype | Non Bt genotype |
| 2012 | 1.3–8.9 (30) | 0.9–8.2 (40) | 0.0–27.1 (33) | 0.7–25.5 (33) | 1.0–46.3 (41) | – |
| 2013 | 2.1–48.0 (31) | 1.1–55.0 (31) | – | 0.7–28.4 (35) | – | 2.9–109.8 (41) |
| 2014 | 0.8–54.9 (37) | 0.4–38.9 (37) | – | 10.0–145.4 (33) | – | 0.0–120.6 (39) |
| 2015 | 0.3–57.7 (31) | 0.2–55.5 (30) | – | – | – | – |
| 2016 | 1.7–17.3 (31) | 2.5–26.9 (32) | – | 0.0–83.9 (40) | – | 0.4–28.6 (38) |
| 2017 | 0.0–45.7 (30) | 0.4–41.3 (30) | 2.3–41.7 (41) | 1.2–46.5 (39) | 3.6–4.1 (30) | 3.2–31.5 (39) |
| 2018 | 5.8–31.2 (29) | 5.9–36.4 (30) | 0.0–23.5 (37) | – | 0.7–10.6 (27) | – |
| 2019 | 3.8–18.4 (31) | 3.9–29.5 (30) | 0.0–28.1 (27) | 0.0–29.5 (29) | 0.1–9.4 (33) | 0.0–8.2 (34) |
| 2020 | 0.7–35.8 (38) | 0.5–48.1 (38) | 4.6–98.4 (38) | 0.0–80.6 (38) | 1.0–41.8 (31) | 1.3–46.5 (31) |
| 2021 | 1.2–13.7 (38) | 0.3–28.0 (29) | 0.5–17.4 (40) | 0.4–38.6 (37) | 1.8–15.2 (33) | 0.8–16.4 (34) |
| 2022 | 6.5–74.1 (29) | 11.1–84.1 (29) | 2.67–111.11 (28) | 4.43–146.22 (28) | 4.50–94.20 | 4.4–105.20 (30) |

Note:
SMW, standard meteorological weeks.

recorded, seasonal mean per three leaves as 5.2, 3.9 and 3.8; in BG-II genotype whereas, 3.8 (2019) and 14.1 (2020) in HS-6 (Table 2). In 2021, the population of whiteflies per three leaves varied from 1.8 to 15.2 in BG-II and from 0.8 to 16.4 in HS-6. However, in 2022, this range expanded to 4.50 to 94.20 for BG-II and 4.4 to 105 for HS-6. Peak population of

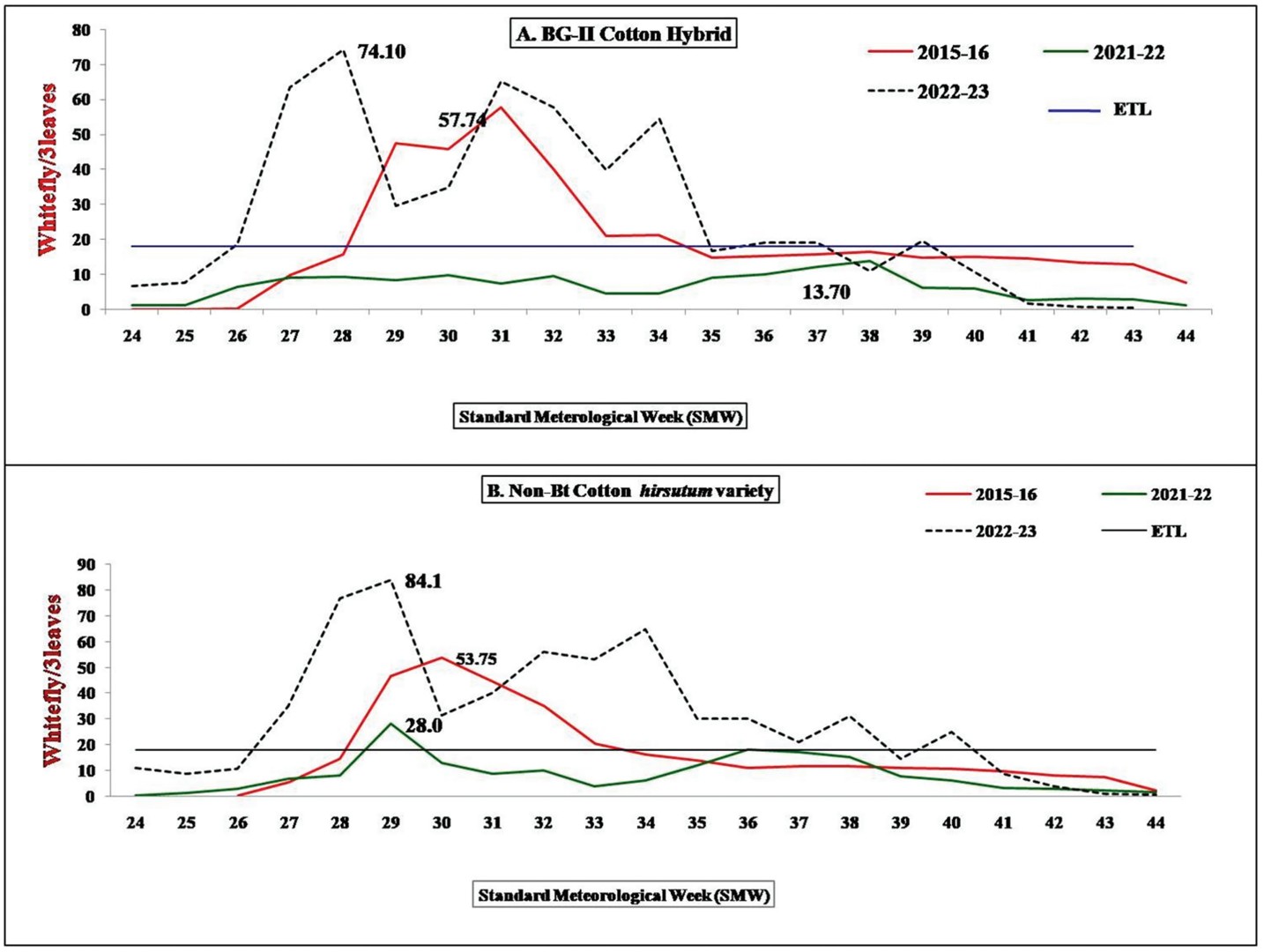

**Figure 1 Comparison of seasonal population dynamics based on experiment under unprotected conditions during 2015 and 2022.** (A) BG-II cotton hybrid; (B) Non-Bt cotton *hirsutum* variety. ETL, economic threshold level.

whitefly during 2022 was recorded during 30[th] SMW. The seasonal mean per three leaves recorded was 11.7 and 25.89 in BG-II and 6.5 and 28.10 in HS-6 during 2021 and 2022. The highest peak seasonal mean of whitefly adults per three leaves was documented during 2022 cotton season.

In Rajasthan state, Sriganganagar has the highest area under cotton cultivation where on farm experiment on population dynamics under unprotected conditions conducted during 2012, 2013 and 2014. For the year 2012 and 2013, whitefly peak incidence was recorded during the 33[rd] and 35[th] SMW while in 2014 it reached it's peak in 33[rd] week. The entire North zone experienced the peak during later part of 2014 season supporting a good start for the next season of 2015. The increasing trend in seasonal mean per three leaves recorded during 2012, 2013, and 2014 was 12.1, 12.8 and 53.9 in HS-6 (Table 2). The population per three leaves in BG-II ranged between 2.3–41.7 during 2017, whereas it

was not cultivated in 2015 and 2016. In HS-6, population per three leaves ranged from 0.0–83.9 (in 2016) and 1.2–46.5 (in 2017) but in 2015, HS-6 was not available for observations. During 2015, 2016 and 2017 peak population of whitefly was recorded between 40th and 41st SMW. Stable seasonal mean per three leaves of 17.7 (in 2016) and 19.1 (in 2017) was recorded in HS-6 (Table 2). During 2018, 2019 and 2020 whitefly per three leaves, ranged between 0.0 to 23.5 with a seasonal mean of 8.9, 0.0 to 28.1 with a mean of 10.2 and 4.6 to 98.4 with a seasonal mean of 35.8, respectively in BG-II genotype. As for HS-6, genotype was not cultivated in 2018, but during 2019 seasonal mean per three leaves was 10.1 and 32.1 during 2020 (Table 2). During 2021 and 2022 seasonal mean (average population range) per three leaves was 5.6 and 52.51 respectively, in BG-II and 9.2 and 47.88 respectively, in HS-6 genotype (Table 2). Peak population of whitefly was recorded during 28th SMW in 2022 season.

The seasonal mean of the whitefly recorded on various genotypes under unprotected conditions clearly indicated that the population of whitefly was significantly higher during 2022 than 2015 season of whitefly outbreak. In case of both genotypes BG-II Cotton hybrid and HS-6, non-*B. hirsutum* genotype seasonal means of whitefly during 2013, 2014, 2016, 2017, 2018, 2019 and 2020 were almost similar. There was significant differences in average seasonal means over the year. The highest seasonal mean recorded in the entire zone from 2012 to 2022 noticed in 2022, surpassing the levels observed in 2015, a year known for whitefly outbreak (Table 3). As far as the severity of whitefly population persistence above the economic threshold level (18 adults per three leaves) during the season is considered, whitefly stayed above ETL for more than 10 weeks during 2022 followed by 2017, 2015 and 2020.

In 2015 the factors responsible for the outbreak were late sowing, bushy as well as hairy genotypes and insecticides mixtures were responsible for the resurgence and was uniformly noticed throughout the zone whereas during 2022, availabilty of alternate host as summer *moong* supported the early initial build up and was replicated as a outbreak in many blocks of the North Zone and was not a resurgence due to insecticides usages.

### Interspecific competition among the sucking pests in the north zone recorded during 2015 to 2022

Out of the 8 years of present surveys at farmers field locations, for 5 years whitefly dominated among the sucking pests followed by thrips for 3 years and leafhoppers for 1 year. During 2015 out of the total 1,089 surveyed locations the whitefly crossed ETL at 56.93 per cent spots, however the corresponding figure during 2022 out of the 1,317 surveyed locations was 44.42%. The minimum locations above ETL in the studied decade was 2.17 per cent of total 2,485 surveyed spots in 2018. The season 2022 was very unusual in terms of pest incidence as 18.38 per cent of the surveyed locations were also found to be above ETL for leaf hopper population, which has was highest in the studied decade. The lowest number of locations (2.36% *i.e.* 55 out of 2,335) recorded above ETL leafhopper population was during 2016. Thrips are known to appear earlier than whitefly and leafhopper, the locations recorded above ETL population of thrips were maximum during 2021 (152 out of 1,220) and minimum (25 out of 2,335 *i.e.* 1.07%) during 2016 (Table 4).

**Table 4  Farmers' field locations surveyed for monitoring of sucking insect pests infestation.**

| Year | Total no. of locations surveyed | Number and percentage of locations crossed economic threshold level | | | | | |
| | | Whitefly | | Leaf hopper | | Thrips | |
| | | Numbers | *Percent* | Numbers | *Percent* | Numbers | *Percent* |
| 2022 | 1,317 | 585 | 44.42[b] | 242 | 18.38[a] | 29 | 2.20[cd] |
| 2021 | 1,220 | 84 | 6.89[d] | 113 | 9.26[b] | 152 | 12.46[a] |
| 2020 | 1,622 | 380 | 23.43[c] | 79 | 4.87[c] | 153 | 9.43[ab] |
| 2019 | 2,125 | 73 | 3.44[d] | 72 | 3.39[cd] | 127 | 5.98[b] |
| 2018 | 2,485 | 54 | 2.17[d] | 92 | 3.70[cd] | 75 | 3.02[bc] |
| 2017 | 1,956 | 125 | 6.39[d] | 82 | 4.19[c] | 79 | 4.04[c] |
| 2016 | 2,335 | 132 | 5.65[d] | 55 | 2.36[d] | 25 | 1.07[d] |
| 2015 | 1,089 | 620 | 56.93[a] | – | – | – | |

**Note:**
Different letters as superscripts within a column indicate significant differences ($p < 0.05$); Means were separated based on the Tukey's honestly significant difference test with $\alpha = 0.05$ level of significance. Farmers' field locations were surveyed in Hisar & Sirsa (Haryana), Bathinda & Faridkot (Punjab and Sriganganagar (Rajasthan) at weekly interval to study the incidence of whitefly and issue advisory to the farmers.

Based on the data from 2015 onwards, during 2022 early whitefly infestation during the season was observed, with the highest seasonal mean among all years and persisting infestation above ETL consistently for more number of SMWs. This required more number of insecticidial sprays to bring down the pest below ETL to avoid the losses. But based on farmers field surveys higher number of locations were recorded above the economic threshold level during 2015 (620 out of 1,089) in comparison to 2022 (585 out of 1,317) indicating that unlike 2015, 2022 proliferation was not uniform in the entire zone wherein only few blocks of Punjab, Haryana and Rajasthan recorded severe infestations of the whitefly. Thus 2022 infestation may be attributed to whitefly initial build up due to early availabilty of alternate host and shifting to cotton later on followed by other factors but not typically insecticides induced resurgence unlike 2015 as carryover of whitefly during 2021 was less as compared to 2014 when outbreak occurred in succeeding year of 2015.

### Correlation coefficient between whitefly and prevailing weather conditions

In the north zone, sowing is completed by 15th May (20th SMW) and generally incidence of insect starts after 1 month of sowing (23rd SMW onwards). The weather factors in general always have an impact on the population of insect pests but whitefly specifically has interactions with multiple facors. Whereas looking at the weather data, weather parameters seems to be contributed more as more favourable weather persist for longer period during 2022 than in year 2015. The regression analysis between prevailing weather conditions and average population of white fly/3leaves was presented as Table S13. The maximum $R^2$ value was obtained in the year 2021 $R^2 = 0.65$ followed by in the year 2013 $R^2 = 0.64$. Single favourable factor may help in initiation of incidence but more than one weather factors alongwith the agronomical parameters especially crop age and stage supports proliferation of the pest. The meteorological data of Sirsa recorded consecutively for 11

**Table 5 Correlation coefficient between whitefly and prevailing weather conditions at ICAR-CICR, Sirsa location.**

| Year | Temperature (°C) | | Humidity (RH%) | | Rainfall(mm) |
|------|------|------|------|------|------|
| | Tmax | Tmin | RH (M) | RH (E) | |
| 2012 | −0.46 | −0.66** | 0.31 | 0.12 | 0.25 |
| 2013 | 0.16 | 0.44 | 0.25 | 0.16 | 0.02 |
| 2014 | −0.46** | −0.01 | 0.62** | 0.61** | 0.38 |
| 2015 | −0.46 | 0.39 | 0.32 | 0.63* | 0.03 |
| 2016 | −0.22 | 0.54* | 0.50* | 0.63** | 0.17 |
| 2017 | −0.20 | 0.48* | 0.47* | 0.56* | 0.25 |
| 2018 | 0.16 | 0.52* | 0.24 | 0.60** | 0.19 |
| 2019 | −0.48* | −0.41 | 0.54* | 0.63** | 0.30 |
| 2020 | −0.28 | −0.06 | 0.29 | 0.09 | 0.13 |
| 2021 | −0.41 | −0.08 | 0.56* | 0.69* | −0.16 |
| 2022 | 0.04 | 0.46* | 0.31 | 0.63** | 0.45 |

**Notes:**
* Significant at 5% level of significance ($p < 0.05$).
** Significant at 1% level of significance ($p < 0.01$).

years has been correlated with the weather parameters where minimum temperature and relative humidity (evening) has more influence on the population of the pest over the years being correlated positively. Rainfall does have a negative impact on the population of pest but it always require higher intensity rainfall which helps in dislodgement of the whitefly nymphs resulting into life cycle break (Table 5; Raw data presented in Table S2 to S12). Due to favorable weather during 2022, second time after 2015, the whitefly has become a major limiting factor for cotton in North zone of India.

## DISCUSSION

Regular observations on the population dynamics of *B. tabaci* on cotton cultivars under unprotected conditions indicated that the 2015 outbreak was not a sudden phenomena but was a result of steady increasing trend of population built up of the whitefly since 2012 in the entire North zone (Haryana, Rajasthan and Punjab) and higher population during 2014 during later part of season has helped in good carry-over of whitefly incidence for the ensuing cotton season of 2015. In 2017, the population level of whitefly was almost similar to 2015 initially but was curbed through the adoption and implementation of timely and effective management strategies. The whitefly populations are mainly regulated by weather factors such as temperature, rainfall, and humidity in general. Both high temperature and humidity correlates positively with the whitefly population build-up (*Pathania et al., 2020*). The data recorded under unprotected conditions for 11 years indicate minimum temperature and relative humidity during evening exhibited positive correlation during majority of the years under study. The similar observation was also recorded by *Sundamurthy (1992)*. Whereas, moderate precipitation and high temperatures are generally favourable for *B. tabaci* leading to its population increases. Furthermore, high intensity rainfall can help in dislodgement of nymphs but low intensity rainfall leading to

suitable humidity and high temperature can flare the whitefly population. Particularly, conditions with dry and hot climates with installed irrigation systems are favourable for *B. tabaci* (*Janu & Dahiya, 2017*). Considering their short generation time, large populations can develop in summer to migrate from other alternate hosts adjoin to cotton fields during *kharif* season crop (*Skendzic et al., 2021*) especially the cotton in north India. The first outbreak of the whitefly in cotton was reported in the late 1920s and early 1930s in Northern India (*Misra & Lamba, 1929*; *Hussain & Tehran, 1933*). Specifically in India the whitefly occurrence and outbreak in all cotton growing zones were reported mainly in Punjab-North zone (1930–43), Andhra Pradesh-South Zone (1984–87), Tamilnadu-South zone, Maharashtra-Central zone and Karnatka-South zone (1985–87), Gujarat-Central zone (1986–87), Haryana, Punjab and Rajasthan-North zone (2015) (*Kumar et al., 2016*).

North India cultivated cotton in 18.0 lakh ha in 2020, 17.06 lakh ha in 2021 and 16.65 lakh ha during 2022 under irrigated condition and experienced two whitefly outbreaks (COCPC 2023). During the 2015, a serious outbreak of whitefly incidence was recorded during August onwards in the cotton growing areas of North zone (Haryana, Punjab and Rajasthan) leading to a heavy loss in crops (*Kumar et al., 2020*). The infestation recorded during 2022 was advanced to July as compared to August in 2015 outbreak where in severity was more during August (Fig. 1; Raw data presented in Table S1). The pest remained above ETL for a longer durations during 2022 (≥10 SMW) as comaperd to 2015 (above ETL for 6 SMW only). The lowest North zone productivity (358.55 kg lint/ha) was reported during 2015 whereas zonal productivity during 2022 was 482.83 kg lint/ha more than 2015 because the attack of the whitefly was restricted to few blocks of the zone. During 2022 the whitefly incidence was not uniform across the zone, which suggested that the infestation in this particular year may be recognized as a result of the whitefly early buildup, including factors other than usages of insecticides, rather than a typical multi factor resurgence observed in 2015.

## Factors for resurgence

Factors responsible for resurgence were categorized as agronomical, climatic and insecticide induced hormoligosis and their mixtures (*Kranthi, 2015*; *Kaur, 2022*). Among cultural factors, sowing beyond recommended date either late or early has become very common practice. In North Zone cotton growing states of India, the normal recommended sowing time is from 15[th] April to 15[th] May of the year. However, based on the availability of canal water and cropping system *i.e.* cotton-wheat or cotton-mustard, the sowing window has been widened year after year. During 2022, sowing commenced from 25[th] March and continued till first week of June (>60 days sowing window) which may have helped in the whitefly proliferation creating sequential niche for the multiplication of the pest. This eventually led to more infestation as well as more adverse effects on the late sown cotton crop. Cotton is highly nutrient intensive crop and requires a balanced nutrition. However, imbalanced fertilizer application, either excessive use of nitrogenous fertilizers or under nutrition in the fear of the whitefly and poor or improper application of phosphatic or potassic fertilizers is very common (*Saleh et al., 2016*). Availability of alternative potential hosts during the season and off-season may have created favorable niche for this

pest. Whereas availability of collateral hosts further support in build-up and subsequent outbreak, especially summer *moong* in Punjab during 2022 was one of the major factors.

The region witnessed interspecific competition among sucking pests but the dominance of the whitefly has well been proved and outbreak of these pests has been observed at frequent intervals (*Kumar et al., 2020*). The whitefly is known to have about 11 generations during cotton growing season (April-October) in North zone and any development with respect to insecticide reaction in the form of resistance can easily be passed on to the next generation. The introduction of 'synthetic pyrethroids' in 1981 acted as initial trigger for the whitefly upserge (*Patil, Nandhihalli & Hugar, 1990*). Usage of some insecticides and their sub-lethal dosages are responsible for hormoligosis. Earlier experiments proved that acephate when sprayed four times at fortnight intervals led to heavy damage from whiteflies leading to plant death in the form resurgence (*Kranthi, 2015*). Tank-mix of pyrethroid+acephate more frequently used in north India led to quick surge in whiteflies resulting in outbreaks. Another insecticide, fipronil recently was found to cause the whitefly resurgence in north India (*Kumar et al., 2019a*, *2019b*) particularly during 2015.

The natural enemies are generally more susceptible to insecticides than insect pests. The generalist predators (coccinellids, chrysopids and soiders) as well as parasites are generally present every where in the plant and more vulnerable to be exposed to insecticides (*Dhaliwal, Singh & Jindal, 2013*). The destruction of natural enemies due to continuous insecticide applicatins even at sub lethal doses also hamper the population of these whitefly population regulators. The whitefly with continuous usage has developed resistance to almost every insecticide employed for its control (*Sharma et al., 2013*). Insecticide resistance is among the major factors responsible for outbreak of the whitefly in 2015. Besides, there were very few effective insecticides available against the whitefly. This has resulted in excessive and indiscriminate insecticide sprays that disrupted ecosystems, which led to severe whitefly outbreaks and further development of resistance. Insecticide resistance monitoring carried out by ICAR- Central Institute for Cotton Research (R Kumar, 2018, unpublished data) showed a high level of insecticide resistance to neonicotinoids (acetamiprid and thiamethoxam) and organophosphate (triazophos and acephate) in use. Even some of the new generation insecticides used very frequently *e.g.*, diafenthiuron, flonicamid *etc.*, are at risk of resistance development by the notorious pest. Among climatic factors, during 2015, rainfall of less than 100 mm up to July coupled with cloudy conditions led to high humidity which created conducive weather for the insect pest. Each factor discussed above favored the whitefly proliferations in the years of upsurged population. Weather in general was drier with low rainfall, lower humidity and higher day and night temperatures. The crop was under stress at many locations due to the prevailing dry weather conditions and prone to attack of the insect-pests.

## CONCLUSION

A severe outbreak of the whitefly was experienced in 2015 in North India on cotton followed by another outbreak during 2022, with a higher seasonal mean during the latter season. The incidence of pests initiated earlier in cotton during 2022 and continued to survive and thrive above ETL for a longer duration as compared to the 2015 outbreak.

Further locations were above ETL during 2015 as compared to 2022 at farmers' field, which indicated that the severity of 2022 outbreak was higher but only in few pockets, unlike the 2015 outbreak, which occurred in the entire north zone. The advancement in infestation and the higher severity of the whitefly during 2022 in contrast to the past few years was attributed to the availability of alternate hosts and susceptible genotypes initially, and later on the inappropriate use as well as application methodology of insecticides; other seasonal agronomic and weather factors collectively triggered the whitefly outbreak in the north zone.

### Plant guideline statement

Experimental research and field studies on plants (either cultivated or wild) was as per the relevant institutional guidelines and legislation of ICAR-CICR, Nagpur, Maharashtra, India. No leave samples were collected and the data was taken from standing crop of farmers' field locations in Hisarand Sirsa (Haryana), Bathinda and Faridkot (Punjab) and Sriganganagar (Rajasthan) with prior permission.

The authors also confirmed that no live animal/parts of the animal were used in the study.

### Funding

The work was conducted under the All India Coordinated Research Project on Cotton. The facilities and funding are provided by Project Coordinator and Head, AICRP on Cotton and Director ICAR-CICR, Nagpur. The funders had no role in study design, data collection and analysis, decision to publish, or preparation of the manuscript.

### Grant Disclosures

The following grant information was disclosed by the authors:
All India Coordinated Research Project on Cotton.
Project Coordinator and Head, AICRP on Cotton and Director ICAR-CICR, Nagpur.

### Competing Interests

The authors declare that they have no competing interests.

### Author Contributions

- Rishi Kumar conceived and designed the experiments, performed the experiments, analyzed the data, prepared figures and/or tables, authored or reviewed drafts of the article, and approved the final draft.
- Satish Kumar Sain performed the experiments, prepared figures and/or tables, authored or reviewed drafts of the article, and approved the final draft.
- Satnam Singh performed the experiments, prepared figures and/or tables, authored or reviewed drafts of the article, and approved the final draft.
- Suneet Pandher performed the experiments, prepared figures and/or tables, authored or reviewed drafts of the article, and approved the final draft.

- Roop Singh Meena performed the experiments, prepared figures and/or tables, authored or reviewed drafts of the article, and approved the final draft.
- Anil Jakhar performed the experiments, prepared figures and/or tables, authored or reviewed drafts of the article, and approved the final draft.
- Jasjinder Kaur performed the experiments, prepared figures and/or tables, authored or reviewed drafts of the article, and approved the final draft.
- Mandeep Pathania performed the experiments, prepared figures and/or tables, authored or reviewed drafts of the article, and approved the final draft.
- Debashis Paul analyzed the data, prepared figures and/or tables, authored or reviewed drafts of the article, and approved the final draft.
- Prakash A.H. conceived and designed the experiments, prepared figures and/or tables, authored or reviewed drafts of the article, and approved the final draft.
- Prasad Y.G. conceived and designed the experiments, prepared figures and/or tables, authored or reviewed drafts of the article, and approved the final draft.

### Data Availability

The raw data is available in the Supplemental File.

### Supplemental Information

Supplemental information for this article can be found online at http://dx.doi.org/10.7717/peerj.17476#supplemental-information.

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
