# Peer review of "Investigating the second whitefly population outbreak within a decade in the cotton growing zone of North India"

_PeerJ, doi:10.7717/peerj.17476_

## Round 0.1 · original submission · Major Revisions

· Academic Editor

Major Revisions

Authors are kindly requested to pay attention to the comments provided by the reviewers, especially those related to data analysis and data presentation. There is room for improvement in the quality of English language usage. I believe the authors will address these concerns when submitting the revised version of the manuscript.

**Language Note:** The Academic Editor has identified that the English language must be improved. PeerJ can provide language editing services - please contact us at [email protected] for pricing (be sure to provide your manuscript number and title). Alternatively, you should make your own arrangements to improve the language quality and provide details in your response letter. – PeerJ Staff

Reviewer 1 ·

Basic reporting

The text of the article is technically correct. The relevant literature is cited. Some minor corrections have been suggested through track changes in the manuscript itself which need to be incorporated before its publication The figures and tables are relevant to the content of the article.

Experimental design

The experimental design is up to the mark.

Validity of the findings

The findings of the present article will help know the underlying causes for the outbreak of whitefly in cotton and will be helpful in sustainable management in the future.

Additional comments

Some suggestions have been given in the manuscript that need to be addressed.

Annotated reviews are not available for download in order to protect the identity of reviewers who chose to remain anonymous.

Reviewer 2 ·

Basic reporting

This manuscript provides useful information on insights about the severity of whiteflies in cotton fields and factors associated with upsurges of the whiteflies in North India. However, there are many items that need attention in a revision. More background of the investigation and novelty of the research should be described sufficiently in the manuscript. To only refer to the subject insect as “whitefly” is not sufficient; there are nearly two thousand species of whiteflies. There are many awkward sentences and many typographical errors that need to be corrected throughout the manuscript. The difference between “unprotected condition” and “unsprayed condition” is not clear. Define what is considered “unprotected conditions”. All abbreviations need to be defined the first time mentioned in the Abstract, text, and Tables.

Experimental design

Numerous fields were sampled across 11 years. The rigor of the experiment only becomes visible when one looks at information in the tables. More details are needed in the Materials and Methods, such as size of plots, cultivars, years and dates of sampling, and the data that were collected.

Validity of the findings

There is valuable seasonal information in this manuscript. Many cause and effect statements were made without supporting data or citations. Additional statements were made as facts, but they lacked citations. In comments about peak populations, are these significantly statistical peaks? Also, because seasonal population data are not shown, a reader is not able to see trends. Statistical analyses were not done for data from all zones even though the experimental design appears to have been th same.

Additional comments

1. Line 1-2; Consider revising the title of the manuscript to: “ Investigating the second whitefly population upsurge within a decade in the cotton growing zone of North India”.
2. Line 4; insert a space to revise “S.K.Sain” to “S.K. Sain”.
3. Line 5; revise “Singh” to “S.”
4. Line 5; insert a space to revise “A.H.Prakah” to “A.H. Prakah”.
5. Lines 6-17; add the country for each affiliation.
6. Line 2; revise to “The whitefly Bemisia tabaci is ….”
7. Line 24; revise to “North India and this was followed by a profound.…”
8. Line 23-24; revise to “outbreak or the result of a multi factor resurgence.”
9. Ln 26 define “ETL” when used the first time.
10. Line 26; revise “whitefly remained” to “whitefly counts remained”.
11. Line 27; revise the comma to a period, and start a new sentence to read “However, during 2022, …..”
12. Line 29; revise “whitefly” to either “the whitefly” or “B. tabaci”.
13. Line 29; insert a space after the comma in “2022,respectively”.
14. Line 30; “longer longer” written twice. Delete one and check the sentence.
15. Line 31; “recorded recorded” written twice; delete one and check the sentence.
16. Line 31; revise “the pest” to “pest”.
17. Line 32; revise “field” to “fields”.
18. Line 32; round “44.42%” to one decimal place.
19. Line 33; because abstracts should be able to stand alone from the rest of the manuscript, do not refer to a “figure” here.
20. Line 33; be consistent on the decimal places for the 2022 (on line 32) 2015 (on line 33) percentages.
21. Line 33; revise “Thus in the 2022” to “Thus, in 2022,”.
22. Line 34-35; indicate that “Punja, Haryana, and Rajasthan” are states in India.
23. Line 35; revise “whitefly” to either “B. tabaci” or “the whitefly”.
24. Line 35; revise “The present study” to “This study”.
25. Line 36; revise “complex of causes such as weather factors” to “complex of factors including weather, …..”
26. Ln 41; delete the comma after whitefly.
27. Lin3 42; revise “whitefly” to “this whitefly.”
28. Line 42-44; provide citations for this second sentence in this paragraph.
29. Line 43; insert a comma after “1905” and revise “assuming” to “attaining”.
30. Line 45; revise “1980s” to “the 1980”.
31. Line 46; insert a space after the period “Dhawan et al. 2007).The”.
32. Line 46; revise “whitefly” to either “B. tabaci” or “this whitefly”.
33. Line 49; revise “also observed” to “also been observed”.
34. Line 50; delete the extra period after “etc.”
35. Line 50-53; provide a citation.
36. Line 51’ revise “of cotton” to “of the cotton”.
37. Line 48-53; consider providing taxonomic names for all species of plants listed.
38. Line 53-55; Suitable weather for this insect is not limited to 27°C and 71% RH. Any temperature and humidity that support the population of this whitefly is suitable. Li et al. (Insects, 2021, 12(3): 198) reported that several studies on different crops have reported an optimal temperature range for B. tabaci growth as 20–33°C (Wang & Tsai, Ann. Entomol. Soc. Am. 1996; 89:375–384, Muñiz, Environ. Entomol. 2001; 30: 720–727). Revise the sentence for clarity and to indicate the optimal temperature range and provide a citation.
39. Line 54, “270C” is not correct. The Celsius degree symbol is appropriate as 27°C.
40. Line 56; revise “Whitefly” to “Bemisia tabaci”.
41. Lines 56-61; provide citations.
42. Line 61; revise “whitefly” to either “this whitefly” or “B. tabaci” or “whiteflies”.
43. Line 66; insert a comma after “other hand”.
44. Line 68; insert a comma after “factors”.
45. Line 73; revise “whitefly” to “B. tabaci”.
46. Line 74-76; provide a citation.
47. Line 75; revise “whitefly” to “this whitefly”.
48. Line 77; revise (Worldover” to “Globally,”.
49. Line 77; revise “whitefly” to “B. tabaci”.
50. Line 85; delete a space after (Mahmood et al. 2022)” and insert a space between the period and “The”.
51. Line 85; revise “biotype” to “cryptic species”.
52. Line 86; revise “whitefly” to “B. tabaci”.
53. Line 87; delete a space “( Table 1)” between open bracket and “Table”.
54. Line 88; revise “recroded” to “recorded”.
55. Line 89; revise “blocks” to “sections”.
56. Line 90; revise “whitefly” to “B. tabaci”.
57. Line 92-93; indicate the location that this study targets.
58. Line 95; revise “whitefly, Bemisia tabaci” to “B. tabaci”.
59. Line 96; revise “studies” to “monitoring”.
60. Line 99; delete the space between “and locations” to revise to “and locations”.
61. Line 103; revise “recroded” to “recorded”.
62. Line 106-108; when was this experiment conducted, and what were the genotypes of cotton in this experiment?
63. Line 106-120; clarity is needed; the “whitefly population” was not counted. The population was sampled.
64. Line 111; revise “period, however” to “period; however,”.
65. Line 116; indicate the size of the plots.
66. Line 116; remove space to revise “( 4 cohort” to “(4 cohort)”.
67. Line 116; delete “regularly”.
68. Line 118, correct “recroded” to recorded.
69. Line 121, delete the space between “field s:” to correct to “fields”.
70. Line 122; revise “( Pungab Haryana and Rajasthan)” to “(Pungab, Haryana, and Rajasthan)”.
71. Lines 122-123; clarify meaning of “assess the ground situations”.
72. Line 123; define “ETL” when used the first time in the text.
73. Line 125; revise “populations” to “counts”.
74. Line 126; revise “populations” to “counts”.
75. Line 127; insert a space after the period between “incidence.Similarly” to revise to “incidence. Similarly”.
76. Line 128; revise “population” to “counts”.
77. Line 134, correct “recroded” to recorded.
78. Line 134; revise “population” to “counts”.
79. Lines 138-139 state that means were subjected to the t-test while lines 141-142 state that means were subjected to the Tukey’s test. Clarity is needed on which means for which statistical test.
80. Line 139; “various quantitative characters” is too general. Which characters are being analyzed here.
81. Line 145; this sentence should be moved to the Discussion section; also, there is more than one species of insects as well as more than one species of whiteflies that exist in the North Cotton Growing Zone in India. This sentence needs a revision for accuracy as well as a citation. Also, the genus is misspelled and there is an excess “only” at the end of the sentence.
82. Line 150; correct “recroded” to recorded.
83. Line 152; remove the bold font for “In Haryana”.
84. Line 156; Table 2 should be presented before Table 3.
85. Line 157; revise “Normal peak activity of whitefly was” to “Peak populations of the whitefly were….”
86. Line 159; delete “later part of the season” because 37 SMW is later than 31 SMW by definition.
87. Lines 159-160; the speculation about the population from 2014 to affect the population in 2015 are not Results; instead, this is more appropriate for the Discussion section.
88. Line 160 and elsewhere in the manuscript; revise “&” to “and”.
89. Lines 163-164; data were not presented to support this speculation about damage suppression in 2017. This speculation should be moved to the Discussion section.
90. Line 165 and elsewhere; revise “mean/3leaves” to “means per three leaves”.
91. Line 169 and elsewhere; revise “activity” to “population”. Data were collected on population counts and not on how active the insects were in the crop.
92. Line 174; delete a space between “season .” to revise to “season.”
93. Lines 178-179; do these data refer to table 2?
94. Line 177-178; this first sentence belongs to the Discussion section and a citation is needed.
95. Lines 180-181; awkward sentence.
96. Line 181; this speculation belongs in the Discussion section instead of the Results section.
97. Line 181-18 3; awkward sentence.
98. Line 195; This says that the Rajasthan district is the largest for cotton production, but line 177 says that the Bathinda district is the largest. They cannot both be the largest. Also, this would be more fitting in the Discussion section, and a citation is needed.
99. Line 214; revise “recroded” to “recorded”.
100. Lines 231-234; this is Discussion content.
101. Lines 237-238; this is Discussion.
102. Line 218; insert a space after the period to revise “2018.The” to “2018. The”.
103. Line 222; correct spelling of “Tharshold” to “Threshold”.
104. Line
105. Line 245; revise “recroded” to “recorded”.
106. Line 279; revise “11” to “eleven”.
107. Lines 281-283; Are there any citations to support the statements on these lines? Contrary to the statements, numerous studies have shown that rainfall or overhead irrigation are detrimental to populations of B. tabaci. Also, “high temperature is” is a vague description. Temperatures are detremental when they are above the optimum for B. tabaci.
108. Lines 288-292; provide citations.
109. Lines 293-295; provide citation.
110. Line 306; remove the space before the comma to revise “buildup ,” to “buildup,”.
111. Line 316-319; This study was not set up to assess the impact of planting on whitefly populations. Therefore, instead of saying that there was a cause and effect, one could say that this “may” have been a factor.
112. Lines 321-323; provide citation.
113. Lines 325-327; the study was not set up to test this. Revise because this is only speculation.
114. Line 330; correct the space and period.
115. Line 330; revise “eleven” to “11”.
116. Line 333-334; provide citation.
117. Lines 337-339; provide a citation about the response to the tank mix of pyrethroid + acephate.
118. Line 339; correct spelling of “isectide”.
119. Lines 341-366; most of this paragraph presents cause and effects without supporting data or citations.
120. Line 369; revise “whitefly” to “whiteflies”.
121. Lines 375-379; Revise because this was not shown. These potential factors are speculation.
122. Check the format of the References for consistency and add a period at the end of each reference.
123. Table 1; insert “Cotton in the” before “North Zone” in the figure title.
124. Table 1; spell out “Govt.”
125. Table 2; define “SMW” in table to allow table to stand alone.
126. Table 3; define “SMW” in table to allow table to stand alone.
127. Table 3; indicate the statistical test in the footnote.
128. Table 3; it is not clear why no statistical analysis is shown for data for the Sri Ganganagar and Bathinda zones.
129. Figure 1; define EIL so that the figure can stand alone. Also, the use of red and green data lines and labels will be problematic for any reader who is red-green color blind.

Reviewer 3 ·

Basic reporting

• Authors have not provided weekly (SMW) population data of whiteflies for the years being compared ie.,for the years 2015 and 2022. When some is comparing, progressive population build-up is more relevant than peaks •In obsence of data of weekly population of 19 weekly as claimed in M&M no conclusion can be drawn . •When some is comparing, progressive population build-up is more relevant than peaks. and hence no concluision can be drawn based on data of only peak point.
Please provide Data required of 1) Population dynamics under unprotected conditions: 2015 & 2022
• 2) Pest survillence across farmers. field s: 2015 & 2022
Table1 which is reported data is not required authors can just cote the reference as thgis not an review article •

Experimental design

Adequate

Validity of the findings

Not prossible to state as , no original data provided

Additional comments

• To many grammatical errors and reportative words , please see few have been highlited

Annotated reviews are not available for download in order to protect the identity of reviewers who chose to remain anonymous.

Reviewer 4 ·

Basic reporting

Title: Investigating the second whitefly population upsurge in the North cotton growing zone of India within a decade.
The author’s efforts to investigate the factor responsible for the upsurge of whiteflies in major cotton-growing areas of India. The experiments laid at different locations and procedures adopted for the survey and surveillance of Whitefly in this study were appropriate. However, I feel that the following observation, data analysis, and interpretation are missing in this Manuscript, and authors must incorporate and address them for further improvement of MS.

Experimental design

The MM section, authors must briefly describes study sites, including latitude, longitude, soil, mean climatic variables and agronomy practices for all unprotected sites. The details of year-wise planting date and observation start date for unprotected sites may also be added in the MM sections
In the survey, the authors counted only adult whiteflies, so why was the nymphal population not considered? Generally, the adult population is considered wherever the trap catches are used for assessing pest population dynamics. But in this study, counts while survey manual count was carried out for recording the whitefly population. Therefore, this needs to be clarified.

Validity of the findings

To interpret the whitefly population upsurge, authors mainly focused on weather parameters and correlation analysis. However, regression analysis needed to be included in this study. Regression analysis is a reliable method of identifying which variables have an impact on whitefly upsurge.
In the result sections in Table 4, i.e. Farmersí field locations surveyed for monitoring of sucking insect pests infestation, only locations crossed ETL and % locations only given. Here, I suggest authors incorporate location-wise/Zone-wise population data by highlighting locations where the population is severe as a supplementary table.
Figures Quality are not Good.
Further, in the discussion, authors used two factors/reasons (Weather and use of insecticides) to interpret the white population upsurge. However, information or data on insecticide usage patter should be given across the surveyed location. If authors recorded this information in the survey, the same may be incorporated; otherwise, discussing pest resurgence does give less meaning.
This needs to be addressed and may be incorporated in the MS.

Additional comments

Overall the Manuscript the Manuscript needs major revision to consider into publishing Peer J
My decision is this Manuscript requires Major Revision

---

## Round 0.2 · accepted · Accept

· Academic Editor

Accept

I expect authors to kindly include the suggestions of the reviewer in their final version of manuscript.

Reviewer 4 ·

Basic reporting

no comment

Experimental design

no comment

Validity of the findings

no comment

Additional comments

Authors made substantial revision. However, in the final MS, I suggests authors to incorporate the statistical values (example r and r2 obtained) in the running text as well. Further, the interspecific competition part may be taken into discussion section.

The manuscript may be ACCEPTED